# High-Resolution Estimates of Fire Severity—An Evaluation of UAS Image and LiDAR Mapping Approaches on a Sedgeland Forest Boundary in Tasmania, Australia

Samuel Hillman [1,*,†] , Bryan Hally [1] , Luke Wallace [2] , Darren Turner [2] , Arko Lucieer [2] , Karin Reinke [1,3] and Simon Jones [1,3]

1 School of Science, RMIT University, Melbourne, VIC 3001, Australia; bryan.hally@rmit.edu.au (B.H.); karin.reinke@rmit.edu.au (K.R.); simon.jones@rmit.edu.au (S.J.)
2 School of Geography, Planning, and Spatial Sciences, University of Tasmania, Hobart, TAS 7001, Australia; Luke.Wallace@utas.edu.au (L.W.); darren.turner@utas.edu.au (D.T.); Arko.Lucieer@utas.edu.au (A.L.)
3 Bushfire and Natural Hazards Cooperative Research Centre, East Melbourne, VIC 3004, Australia
* Correspondence: samuel.hillman@student.rmit.edu.au
† Current address: School of Science, RMIT University, Melbourne, VIC 3001, Australia.

**Abstract:** With an increase in the frequency and severity of wildfires across the globe and resultant changes to long-established fire regimes, the mapping of fire severity is a vital part of monitoring ecosystem resilience and recovery. The emergence of unoccupied aircraft systems (UAS) and compact sensors (RGB and LiDAR) provide new opportunities to map fire severity. This paper conducts a comparison of metrics derived from UAS Light Detecting and Ranging (LiDAR) point clouds and UAS image based products to classify fire severity. A workflow which derives novel metrics describing vegetation structure and fire severity from UAS remote sensing data is developed that fully utilises the vegetation information available in both data sources. UAS imagery and LiDAR data were captured pre- and post-fire over a 300 m by 300 m study area in Tasmania, Australia. The study area featured a vegetation gradient from sedgeland vegetation (e.g., button grass 0.2 m) to forest (e.g., *Eucalyptus obliqua and Eucalyptus globulus* 50 m). To classify the vegetation and fire severity, a comprehensive set of variables describing structural, textural and spectral characteristics were gathered using UAS images and UAS LiDAR datasets. A recursive feature elimination process was used to highlight the subsets of variables to be included in random forest classifiers. The classifier was then used to map vegetation and severity across the study area. The results indicate that UAS LiDAR provided similar overall accuracy to UAS image and combined (UAS LiDAR and UAS image predictor values) data streams to classify vegetation (UAS image: 80.6%; UAS LiDAR: 78.9%; and Combined: 83.1%) and severity in areas of forest (UAS image: 76.6%, UAS LiDAR: 74.5%; and Combined: 78.5%) and areas of sedgeland (UAS image: 72.4%; UAS LiDAR: 75.2%; and Combined: 76.6%). These results indicate that UAS SfM and LiDAR point clouds can be used to assess fire severity at very high spatial resolution.

**Keywords:** photogrammetry; UAS; LiDAR; 3D remote sensing; vegetation; RPAS; drone; structure; fuel structure; fire severity

## 1. Introduction

Many of the world's ecosystems have co-evolved with specific regimes of fire [1–4], which includes the frequency, extent, season, intensity and subsequent severity of fire. Fire severity is a critical element of the fire regime because it can predicate the ecosystem response [5]. Fire severity was quantitatively defined by Keeley [6] as the change in vegetative biomass following fire. In the broader literature, measures of severity are informed by change indicators such as crown volume scorch, percentage fuel consumption and tree mortality [7–11].

Fire severity assessments can be completed using techniques ranging from traditional field-based visual assessments through to established and emerging remotely-sensed assessments. Remote sensing methods that measure fire severity have typically used passive sensors to capture imagery from satellite or fixed-wing platforms [12–15]. Satellite sensors provide large area coverage and can generally capture a complete view of large wildfires with the benefit of lower associated costs [12,16]. Satellite sensors are limited by the frequency of observations and the spatial resolution of the sensor in categorising fire severity. In contrast, fixed wing aerial capture has greater flexibility in deployment for capturing on-demand imagery with higher spatial resolution, albeit at significantly higher cost. Fire severity classifications have been derived from single-date and multitemporal captures using spectral indices [12,17,18]. Indices are generally selected to be sensitive to the changes in vegetation health and condition often caused by fires [19–22]. A threshold at the sampling resolution of the sensor can be implemented to characterise fire severity classes for field validation or aerial photo interpretation. It should be noted that aerial photo interpretation can be completed independently of spectral index implementation [23].

High-resolution imagery captured using unoccupied aircraft systems (UAS, also referred to as drones or unmanned aerial vehicles (UAVs)) have been used in conjunction with supervised classifications (an algorithm which learns on a labelled dataset and can evaluate its accuracy on training data) to map fire severity [24–29]. Image capture from UAS presents a potential improvement in temporal and spatial resolution over airborne and satellite sensors for small areas, e.g., at several hectares to square kilometres. UAS imagery has been used previously to monitor vegetation health and condition, forest condition, soil conditions and ecological planning [30–33]. High-resolution pre- and post-fire imagery has been used to derive difference burn ratios [24,34,35]. For example, McKenna et al. [24] applied the Excess Green Index, Excess Green index Ratio and Modified Excess Green Index to derive fire severity maps with results comparable to multispectral satellite data using difference NVDI and difference NBR [21]. Arkin et al. [25] achieved an accuracy of 89.5% ± 1.5% at 5 m and 85.4% ± 1.5% at 1 m when applying a supervised classification to post-fire UAS imagery, employing textural and structural metrics as predictor variables to produce fire severity and land cover maps.

UAS LiDAR systems provide a means to collect high-resolution 3D data. The high density data collected from UAS platforms have been used to derive metrics of tree height, canopy and density [36–39]. Recently, UAS LiDAR has also been used to detect fine-scale vegetation which would contribute to fire behaviour beneath the canopy [40] with the active sensor allowing for penetration through the canopy to resolve below-canopy vegetation. The applicability of this technology to detect structural change has predominantly been used in forestry contexts [41,42]. Jaakkola et al. [41] demonstrated the ability of UAS LiDAR to detect changes within the canopy which were altered through physically removing branches and leaf material. Wallace et al. [42] produced similar results with UAS LiDAR point clouds successfully showing change from pruning in a Eucalyptus stand.

Limited studies have investigated the link between multi-temporal vegetation structural characteristics and assessing fire severity from wildfire [43–46]. Prior research has shown the utility of UAS point clouds to measure disturbance, and UAS imagery to measure fire severity [24–29]. To the authors' knowledge, there have been no studies that evaluated pre- and post-fire UAS LiDAR variables to map land cover and fire severity across a sedgeland–forest boundary. There is an unresolved debate about the importance of fire, soil or both factors in maintaining these boundaries [47]. The objective of this study was to evaluate the effectiveness of structural metrics derived from UAS LiDAR for predicting fire severity. The first stage of the study applied a supervised classification to pre-fire UAS imagery and UAS LiDAR variables to map land cover. The second stage of the assessment classified fire severity within each land cover class to map fire severity across the study area. The study provides a comparison of accuracy between image-only, LiDAR-only and combined LiDAR and image predictor variables for mapping land cover and fire severity.

## 2. Materials and Methods

### 2.1. Study Area and Fire

The Weld River study area is located approximately 50 km southwest of Hobart in Tasmania, Australia Figure 1. The study area consists of a 300 m × 300 m plot that captures a sedgeland forest boundary; vegetation types vary from *Gymnoschoenus sphaerocephalus* (button grass) plains in the north of the plot to *Melaleuca squamea* and *Eucalyptus nitida* approximately 4 m high in the intermediate zone, grading to a tall forest that at this site is dominated by *Eucalyptus obliqua* and *Eucalyptus globulus*. The dominant understorey species within the tall forest were *Monotoca glauca* and *Pomederris apetela*. There are significant variations in topography, ranging from 40 m to 68 m above mean sea level, with gullies present throughout the study area. The Weld River bisects the southwest corner of the study area.

Pre-fire datasets were acquired in September 2018. Following this data acquisition, a wildfire (Riveaux Road fire complex) occurred in January 2019 [48]. Post-fire datasets were acquired in May 2019.

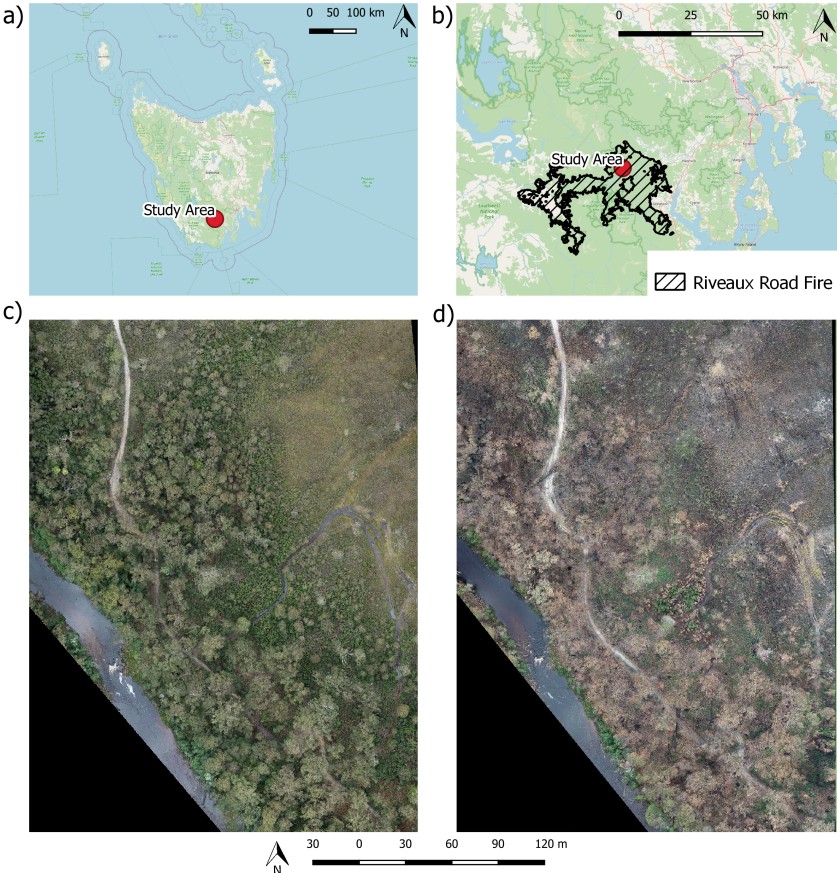

**Figure 1.** (**a**) The location of the site in Tasmania, Australia; (**b**) the location of the Riveaux Road Fire; (**c**) an image of the study area plot captured before the Riveaux Road fire complex (September 2018); and (**d**) an image of the study area plot captured post fire (May 2019).

### 2.2. Data Collection and Pre-Processing

#### 2.2.1. Ground Control

To co-register the data derived from the respective sensors pre- and post-fire, ten Propeller Aeropoints ground control targets were distributed throughout the plot at locations that provided clear-sky views and allowed for a strong network geometry. The position of each target was calculated through the onboard GNSS receiver and post-processed against base stations from a Continually Operating Reference Station network. A base station was also set up on the northwestern edge of the plot, which remained running for the duration

of both surveys—approximately 5 h each time. This base station was used to provide correction information for the positioning unit integrated in the UAS LiDAR system.

### 2.2.2. UAS LiDAR

LiDAR data were captured with two separate sensor systems pre- and post-fire event. Pre-fire data were captured with a custom-built UAS developed at the University of Tasmania, Australia. This system consisted of a DJI M600 platform, a Velodyne Puck (VLP-16) LiDAR scanner and an Advanced Navigation Spatial Dual coupled GNSS and IMU sensor. The VLP-16 scanner features 16 scan layers with a 30° vertical Field Of View (FOV), which equates to a 15° forward and backward distribution of the scan lines in the flight direction (+15° to −15° from nadir) and scan lines that are separated by approximately 2°. A maximum of two laser returns per pulse are collected with 300,000 pulses per second for the full 360° view of the scanner. The scan angle was limited to −40° to +40° in the across-track direction (80° field-of-view) resulting in approximately 60,000 pulses per second. The scanner has a horizontal beam divergence of 0.18° (3 mrad) and a vertical one of 0.07° (1.2 mrad). Data were processed using in-house software developed specifically for the University of Tasmania UAS LiDAR system which has also been in use for the production of point clouds in [49–51].

Post-fire data were captured with a RIEGL miniVUX-1 LiDAR scanner integrated with the APX-15 IMU sensor onboard a DJI M600 platform. The miniVUX-1 is a rotating mirror scanner with a 360° FOV. A maximum of five returns per pulse are collected with 100,000 pulses per second and a beam divergence of $1.6 \times 0.5$ mrad. Data were processed using the RIEGL UAS workflow by firstly adjusting the trajectory of the flight lines using the on-board IMU and GNSS with local corrections using POSPac software. The trajectory of the flight lines was then adjusted in RiProcess with segments of the flight lines trimmed to cover the plot and scan angles reduced to the same parameters as the pre-fire dataset. Lastly, point clouds were then extracted to LAS format and merged in CloudCompare v2.12 [52].

The flying height and flight pattern were identical between the two captures with flights completed 20 m above the highest canopy element and the overlap between flight strips being approximately 50%. Both point clouds were filtered to only include first returns.

### 2.2.3. UAS SfM

Images were captured using a DJI Phantom 4 Pro using the integrated RGB camera, which has an 8.8 mm nominal focal length and a 25 mm CMOS 20 megapixel sensor with $2.41 \times 2.41$ µm nominal pixel size [53]. The UAS was flown at a flying height of 60 m above ground level. Nadir imagery was captured within two separate flights with 90% forward and sidelap. Due to changes in lighting conditions across the plot, camera settings were manually set to balance exposure of captured surfaces. This meant that, while the flight path was the same pre- and post-fire, the camera settings used were different (pre-fire: ISO-400, shutter speed 1/500 s and f 3.2; and post fire: ISO-320, shutter speed 1/400 s and f 2.8).

Images were downloaded from the UAS and processed to form a point cloud using Agisoft Metashape Professional v1.5.0 (www.agisoft.com (accessed on 3 January 2021)) software [54]. A sparse point cloud was generated using the high-quality alignment setting where common features were found within the image set. Images were then aligned based on an iterative bundle adjustment to estimate the 3D positions of the matched features. Ground control targets were then identified within the images to georeference the point clouds, in-turn facilitating direct comparison to point clouds derived from laser scanning. The high-quality setting and mild depth filtering were then applied to generate a dense point cloud. Finally, ortho photos with a ground sampling distance of 0.1 m were created within the Metashape software. Manual noise removal was completed to remove spurious points beneath the ground.

The RGB colour-space was then converted to LAB space. The L*a*b*, or CIELab, color space is an international standard for colour measurements and was preferred over RGB space due to the stronger differentiation of red and green space [55]. L is the luminance or lightness component, which ranges from 0 to 100, and parameters A (from green to red) and B (from blue to yellow) are the two chromatic components, which range from $-120$ to 120 [56].

### 2.2.4. Reference Data

To generate reference data for the model a desktop assessment of vegetation type and severity was completed utilising a similar methodology to those described by McKenna et al. [24] and Arkin et al. [25]. The plot was first tiled into 10 m $\times$ 10 m squares. Within each tile two randomly generated points were assigned. These points were split into two unique collections consisting of approximately 250 points in each collection and given to two separate groups of assessors. Each group of assessors consisted of 3 people. For each point, a visual assessment of the orthophoto was undertaken to determine the vegetation class and severity. Once all assessments had been completed, points were summarised to form a final training dataset for each collection of points. A point was included in the training dataset if two or more assessors had agreement on the vegetation class and severity assessment. Once the training dataset was finalised, a spatial join was completed to assign the assessed vegetation and severity value to a segment. Two stages of random forest (RF) classifier were run to emulate the process which assessors completed: first to develop a vegetation classification using only metrics derived from pre-fire products and subsequently in the assessment of fire severity.

### 2.3. Data Co-Registration
Pre to Post Point Clouds

Point clouds were first clipped to ensure the same geographic area was being analysed and compared (this included removing the watercourse and all areas south of the river from both point clouds). Datasets were aligned using a two-step process. The first utilised the position information collected using the on-board position and orientation sensors. GNSS data were post-processed using software systems designed for the respective platforms/control targets. A second stage of alignment was completed through ground surface matching in open areas on rocks and road features. Care was taken to focus upon matching in areas that were likely to be undisturbed by the fire, due to likely structural deformation/slumping of the surface in fire-impacted areas of the plot.

### 2.4. Point Cloud Processing

For post-fire datasets, ground points were identified in the UAS LiDAR and UAS image-based point clouds using the Cloth Simulation Filter (CSF) outlined in Serifoglu Yilmaz et al. [57]. In order to parameterise the CSF filter several areas that were easily identifiable as ground were extracted from the post fire datasets. The filter was optimised by minimising RMSE between the reference bare ground and generated surface (resulting in a Cloth resolution (m) of 0.1 m, a class threshold of 0.05 m, a rigidity of 1, time step of 0.5 and 1000 iterations). Once identified, the ground points were processed to form a Triangular Irregular Network (TIN). The height of the TIN facet at the centre of each cell was then attributed to a 0.02 m Digital Terrain Model (DTM).

The point cloud was normalised based on each point's height above the DTM, thereby providing a representation of the point cloud in relation to the ground. The point cloud was normalised in density using a 0.02 m voxel size in order to account for differences in point density across the plot.

For pre-fire LiDAR and SfM point clouds where there was minimal bare earth to optimise ground filter settings, the ground surface was taken from the post-fire dataset and used to normalise both the pre- and post-fire datasets. An assumption was made that the

ground surface derived from the post-fire point clouds was more accurate than the surface which could be derived from the pre-fire dataset.

Finally, a Canopy Height Model (CHM) was created with the same resolution (0.10 m) and extent as the imagery. Each cell in the CHM was attributed with the above ground height of the highest point that fell within its boundary. As none of the point clouds contained points for every cell, interpolation was undertaken to fill in the missing cells. A Gaussian smoothing kernel ($\sigma$: 1.2) was applied to the entire CHM. Neighbouring missing cell values were ignored in the calculation of the central kernel value. This smoothed version of the CHM was then used to fill where missing pixels in the original version occurred.

### 2.5. Fire Severity Classification

A workflow consisting of area segmentation, segment description and classification was used to generate vegetation and fire severity classification maps. This workflow aimed to divide the study area into four vegetation classes: forest (areas with tall trees and greater than 30% cover), sedgeland, water and bare earth (Table 1). Within each vegetated cover class, the workflow also aimed to encompass three levels of fire severity. The levels of fire severity follow McKenna et al. [24] and are described in Table 2.

**Table 1.** Descriptions of vegetation classification.

| Vegetation Class | Definition | Example Species |
|---|---|---|
| Forest (tall) | Vegetation greater than 3 m in height | *Eucalyptus obliqua, Eucalyptus globulus* |
| Sedgeland (short) | Vegetation beneath 3 m in height | *Gymnoschoenus sphaerocephalus, Melaleuca squamea, Eucalyptus nitida* |
| Non-vegetation | Water and Bare earth | N/A |

**Table 2.** Descriptions of fire severity classification based upon land cover classifications.

| Impact | With Forest Vegetation Present | With Sedgeland Vegetation Present |
|---|---|---|
| Severe | >50% crown scorch | Grass combusted (>80 %) exposing bare soil, white or black ash |
| Not-severe | <50% crown scorch | Patchy burn on grass and litter incomplete |
| Unburnt | Unburnt | Unburnt grass, or unchanged conditions |

To facilitate comparison between sensors, the workflow implemented here was completed for three streams of input data: LiDAR-only, image-only and a combined stream (Table 3). For the LiDAR-only and image-only stream, only data available from that sensor were used at each step, whilst in the combined stream the segmentation of the data was based on the ortho image and all features from both the LiDAR and imaging sensors were included in the workflow.

#### 2.5.1. Segmentation

A superpixel approach aggregates regions of similar pixels [58]. Superpixels are often used to capture redundancy in the image and reduce the complexity of subsequent large image processing tasks [58]. The Simple Linear Iterative Clustering (SLIC) algorithm implementation in scikit-image was used [59]. The RGB pre-fire image and canopy height model (aligned to the same grid) generated from the pre-fire LiDAR capture were used as separate inputs into the SLIC segmentation algorithm.

**Table 3.** Segmentation description and metric sources for each of the three processing data streams.

|  | Stream 1—Image-Only | Stream 2—LiDAR-Only | Stream 3—Combined |
|---|:---:|:---:|:---:|
| Segmentation | Pre-image | Canopy Height Model (CHM) | Pre-image |
| Ortho image metrics | ✓ |  | ✓ |
| Ortho image texture metrics | ✓ |  | ✓ |
| Point cloud metrics—UAS SfM | ✓ |  | ✓ |
| Point cloud metrics—UAS LiDAR |  | ✓ | ✓ |
| CHM texture metrics—UAS SfM | ✓ |  | ✓ |
| CHM texture metrics—UAS LiDAR |  | ✓ | ✓ |

The SLIC segmentation algorithm performs K-means clustering on the image data. The number of seeds was kept consistent between the two input data-sources. The number of segments was derived from the area requirement to be approximately the same size as the segments used for the manual validation ($3.14 \, \text{m}^2$). This size was consistent with prior studies that also used validation plots for training a random forest classifier [25] and was also deemed large enough to be able to determine vegetation classification and severity. The compactness and sigma parameters were optimised visually to provide segments consisting of only a single vegetation class and reduce slithers and sharp angles (image: compactness = 20, sigma = 5; LiDAR CHM: compactness = 22, sigma = 10). These settings resulted in a mean area of $3.19 \pm 0.47 \, \text{m}^2$ for the image-derived segments and $3.21 \pm 0.35 \, \text{m}^2$ for the CHM-related segments.

### 2.5.2. Image-Based Features

For each segment derived from the imagery pre- and post-fire and CHM, several descriptors were calculated based on the ortho image and the CHM (Table 4).

For each segment, the means of L, A and B components were calculated. Additionally, the LAB space has been shown to provide stronger severity delineation of vegetation elements in comparison to RGB imagery [60,61].

A further technique to differentiate between regions within the study area was implemented to analyse the texture of the ortho image and CHM. Gonzalez et al. [62] described texture as measures of smoothness, coarseness and regularity of an image region which can be calculated by using structural or statistical techniques. The Grey Level Co-occurrence Matrix (GLCM) method [63] was utilised in this study to describe the texture features within each segment. As per Kayitakire et al. [64] and Rao et al. [65], six texture features were extracted describing angular second moment (ASM), contrast, variance, homogeneity, correlation and entropy. Similar to Gini et al. [66], the GLCM calculations were performed only on one channel (L Channel), to reduce data redundancy. The difference between pre- and post-fire reflectances of the the respective L, A, B and texture variables were generated to be used as predictor variables.

### 2.5.3. Point Cloud Features

Structural properties were extracted from the point cloud for each segment and adjacent neighbours (Table 4). Point cloud properties were extracted for the segment and the respective neighbours to reduce the chances for a segment to be misclassifed (e.g., a segment that fell in a canopy gap may be misclassified as a segment with sedgeland vegetation). The area was clipped out of the point cloud and the Wilkes et al. [67] algorithm was applied to calculate the number of layers and layer location above 0.1 m. The parameterisation of this model utilised the default settings ($\alpha$: 0.3). The vertical distance between the first and second layers was also calculated.

Percentile heights were calculated (10th, 50th and 90th percentile) within each of the segments. The total volume of points within each segment was also calculated. Difference metrics were also calculated between each of the respective structural variables pre- and post-fire.

**Table 4.** Metrics derived from image-based and point cloud products for classification of vegetation and fire severity.

| Image Based Metrics | Image Stream Bands | LiDAR | Description |
|---|---|---|---|
| Mean | LAB | N/A | Metric of each band calculated separately within the segment |
| ASM<br>Contrast<br>Correlation<br>Sum of squares: variance<br>Homogeneity<br>Entropy | L | CHM | Texture calculated from single channel lightness (L) image within segment |
| **Point Cloud Metrics** | | | |
| Percentiles (10th, 50th, 90th)<br>Number of layers<br>Distance between 1st and 2nd layer<br>Volume of points<br>Difference in percentile heights<br>Difference in number of layers<br>Difference in volume | RGB point cloud | LiDAR point cloud | Analysis was conducted for the segment and 2nd level of adjacency to the central segment |

### 2.5.4. Random Forests Classification

A RF classifier was used to investigate the relationship between image, texture and structural metrics with vegetation and severity classification. This model was deemed appropriate to have good predictive capacity without overfitting data with RF classifiers being used in ecological studies previously for classification of discrete severity types [24–26,68–70].

The RF classifier used 1000 trees, splitting one set of the assessment data and associated metrics randomly into 70% training segments and 30% test segments. The training and test data segments were kept consistent across all streams of processing. Data inputs varied depending on the classification. For the vegetation classification, predictor variables were taken from pre-fire datasets. In contrast, the severity assessment utilised pre- and post-fire predictor variables. We implemented a feature selection method that firstly removed correlated variables (>0.75 ) and secondly conducted a Recursive Feature Elimination (RFE) process to determine the optimum set of predictor variables from the initial selection. RFE utilises a backward selection of predictors by firstly building a model on the entire set of predictors and computing an importance score and support for each predictor [71,72]. The least important predictor is then removed, the model is re-built and importance scores are computed again. A consideration when running a RFE is determining the optimum number of features. The optimum number of features was calculated by beginning the loop with all features and progressively removing the least important feature in the dataset. The optimum model was selected based of the highest overall accuracy on the test data. The remaining set of assessment data was used as validation of the model.

The vegetation classification was completed first to identify the vegetation features at each site. The RF classifier was subsequently run to classify the severity of segments that were assigned the land cover class of 'forest' and separately to classify the severity of segments that were assigned the land cover class of 'sedgeland vegetation' (as defined in Table 1).

The results of the RF classification were summarised based on the accuracy of the test data from assessment group 1 and the complete group of assessment 2, using confusion matrices from the *RandomForestClassifer* within the Scikit-Learn Python package [73]. A vegetation and severity classification map was produced to show the classification across the plot. The user and producer accuracies were calculated for each of the data streams for the vegetation and severity classification [74]. To capture the difference in errors made by the models, the McNemar's test was completed between each of the three models.

McNemar's is a nonparametric test based on standardised normal test statistic calculated from error matrices of the two classifiers as follows (Equation (1)) [75–77].

$$Z = \frac{n_{0_0} - n_{0_1}}{\sqrt{(n_{0_0} + n_{0_1})}} \tag{1}$$

where $n_{0_0}$ denotes the number of samples that are misclassified by the first RF model but correctly classified by second RF model and $n_{0_1}$ denotes the number of samples that are correctly classified by second RF but misclassified by the second RF model. The Z value could be referred to the tables of chi-squared distribution with one degree of freedom [78]. McNemar's test can therefore be expressed using a chi-squared formula computed as follows:

$$X^2 = \frac{(n_{0_0} - n_{0_1})^2}{n_{0_0} + n_{0_1}} \tag{2}$$

If the statistic $X^2$ estimated from Equation (2) is greater than a chi-squared table value of 3.84 at 5% level of significance, it implies that the models perform significantly different.

## 3. Results

### 3.1. Vegetation Classification

Vegetation maps produced by each of the three processed data streams (Section 2.5) demonstrated the area classified as forest and sedgeland varied by no more than 2% Figure 2. The combined stream classified 54.9% of the study area as forest and 43.9% as sedgeland, in comparison to 54.1% as forest and 42.5% as sedgeland for the image-only stream and 53.2% as forest and 44.2% as sedgeland for the LiDAR-only data stream.

A similar overall classification accuracy was achieved by all data streams (Tables 5–7). This is also indicated by McNemar's test, which showed no significant differences in the performance of each stream ($p > 0.5$). Furthermore, Producer's and user's accuracy for the classification of forest and sedgeland areas were within 10% of each other across all three data streams (Tables 5–7).

The correlation removal and RFE approach resulted in eight predictor variables being used in the image-only stream, five predictor variables being used in the LiDAR-only data streams and ten predictor variables used in the combined stream (see Appendix A). In the image-only stream, structural variables (describing the 90th percentile height and distance between the top two layers), image variables (describing the *LAB_A* mean and *LAB_B* mean) and texture metrics derived from the CHM (homogeneity and entropy) and ortho image (contrast and correlation) were all used. The LiDAR-only stream also used variables describing structure (layer count and 10th, 50th and 90th percentile heights) as well as the texture metrics (correlation and homogeneity) derived from the CHM. The combined stream used a greater number of variables utilising structure and texture variables derived from both the SfM and LiDAR point clouds and CHM, respectively, as well as variables derived from the ortho image.

**Table 5.** Confusion matrix for image-only data stream describing vegetation classification.

| | Class | Bare Earth | Forest | Sedgeland | Water | User's Accuracy |
|---|---|---|---|---|---|---|
| | | **Reference Data** | | | | |
| **Classified Data** | Bare Earth | 1 | 0 | 1 | 0 | 50.0% |
| | Forest | 2 | 126 | 13 | 1 | 88.7% |
| | Sedgeland | 1 | 30 | 84 | 4 | 70.6% |
| | Water | 0 | 0 | 0 | 5 | 100.0% |
| | Producer's Accuracy | 25.0% | 80.8% | 85.7% | 50.0% | Overall: **80.6%** |

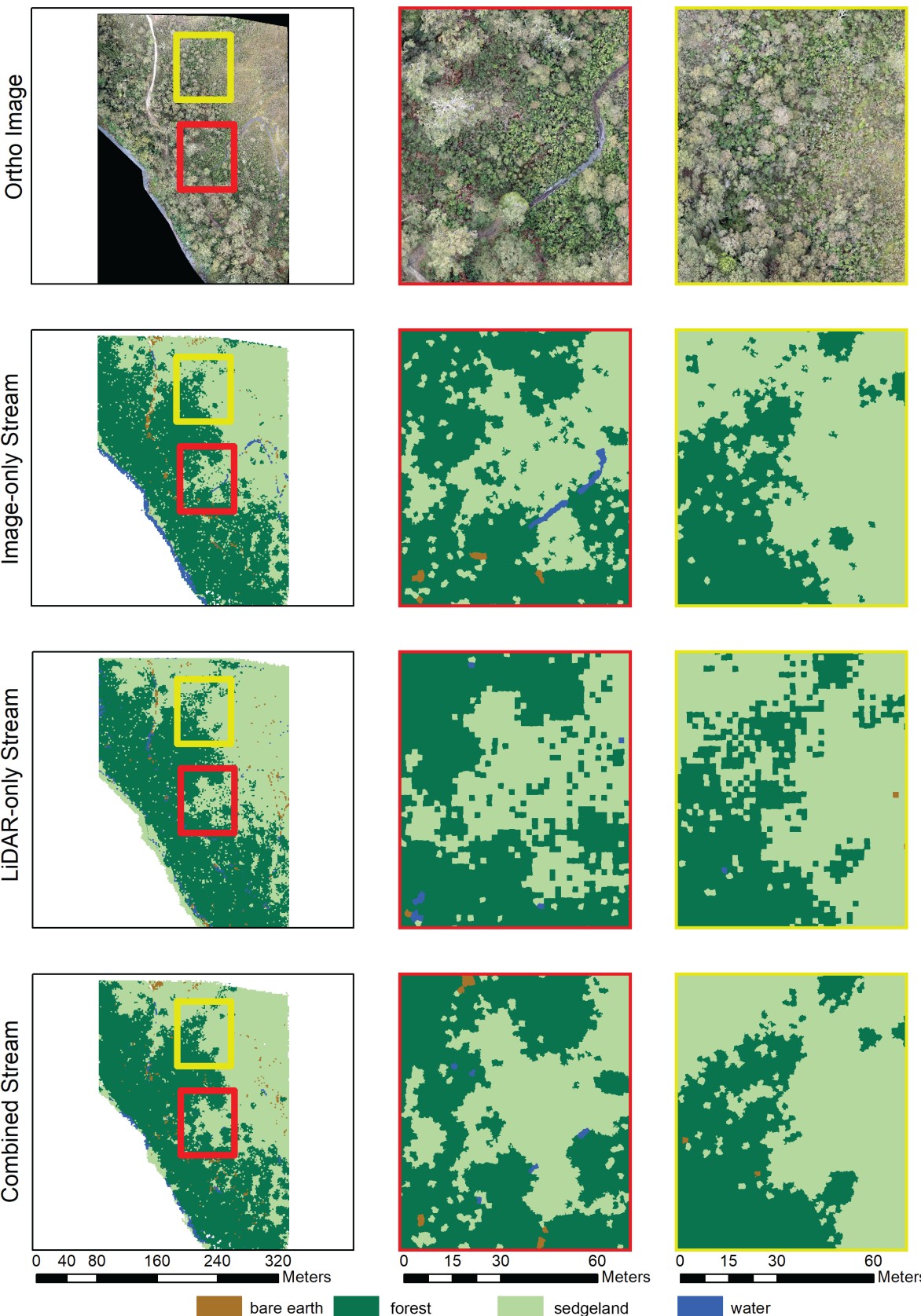

**Figure 2.** Ortho images of the study area and two focused areas. Maps demonstrating the vegetation classification of the image-only data stream, LiDAR-only data stream and combined data stream.

**Table 6.** Confusion matrix for LiDAR-only data stream describing vegetation classification.

| | | Reference Data | | | | |
|---|---|---|---|---|---|---|
| | **Class** | **Bare Earth** | **Forest** | **Sedgeland** | **Water** | **User's Accuracy** |
| **Classified Data** | Bare Earth | 1 | 0 | 1 | 0 | 50.0% |
| | Forest | 1 | 128 | 12 | 4 | 88.3% |
| | Sedgeland | 2 | 30 | 85 | 7 | 68.5% |
| | Water | 0 | 1 | 0 | 3 | 75.0% |
| | Producer's Accuracy | 25.0% | 80.5% | 86.7% | 21.4% | Overall: **78.9%** |

**Table 7.** Confusion matrix for Combined data stream describing vegetation classification.

| | | Reference Data | | | | |
|---|---|---|---|---|---|---|
| | **Class** | **Bare Earth** | **Forest** | **Sedgeland** | **Water** | **User's Accuracy** |
| **Classified Data** | Bare Earth | 0 | 1 | 1 | 0 | 0.0% |
| | Forest | 1 | 131 | 10 | 2 | 91.0% |
| | Sedgeland | 3 | 24 | 87 | 3 | 74.4% |
| | Water | 0 | 0 | 0 | 4 | 100.0% |
| | Producer's Accuracy | 0.0% | 84.0% | 88.8% | 44.4% | Overall: **83.1%** |

*3.2. Fire Severity Classification*

The predominant differences between the severity maps were observed within areas of unburnt riparian vegetation and in areas of vegetation experiencing a green flush post fire (Figure 3). This resulted in small differences in the total area that were classified as unburnt (combined: 2.4%; image-only 3.0%; and LiDAR-only 0.9% (Figure 3), not-severe (combined: 10.7%; image-only: 13.9%; and LiDAR-only: 11.1%) and severe (combined: 84.7%; image-only 79.8%; and LiDAR-only 84.5%). McNemar's test highlighted that the streams that featured predictor variables derived from image products (image-only and combined streams) had similar classification errors (chi-squared; $X^2 = 0.88$). However, a McNemar's test demonstrated differences in performance between the LiDAR-only stream and image-only stream and combined streams (Image-only and LiDAR-only: chi-squared, $X^2 = 4.89$; Combined stream and LiDAR-only: chi-squared, $X^2 = 9.28$).

3.2.1. Classification of Severity within Sedgeland Segments

In areas of sedgeland, the combined stream produced the highest overall accuracy compared to the reference data set (76.6%) followed by the LiDAR-only and image-only data streams (LiDAR: 75.2%; and Image: 72.4%). All data streams had higher producer's and user's accuracy for the severe reference segments in comparison to the non-severe reference segments (Tables 8–10). The highest producer's and user's accuracy for unburnt areas was observed in the image-only stream.

The feature selection approach resulted in 14 predictor variables being used in the image-only stream, 16 predictor variables used for the LiDAR-only stream and 24 predictor variables used for the combined data stream (Appendix B). Using the given training data, variables describing both the pre- and post-fire condition were used in the RF classifier. The feature selection approach when applied to image-only stream resulted in eight variables derived from post-fire capture, five variables describing the difference between pre- and post-fire captures and one variable from the pre-fire capture used in the determination of severity in areas that were classified as sedgeland. These features selected from the image-only stream are in contrast to the feature selection of the LiDAR-only stream that resulted in six variables from pre-fire capture, three variables from post-fire capture and seven variables describing the difference between pre- and post-fire captures. The feature selection approach when applied to the combined stream resulted in ten variables derived

from post-fire capture, four variables derived from pre-fire capture and ten variables describing the difference between between pre- and post-fire captures.

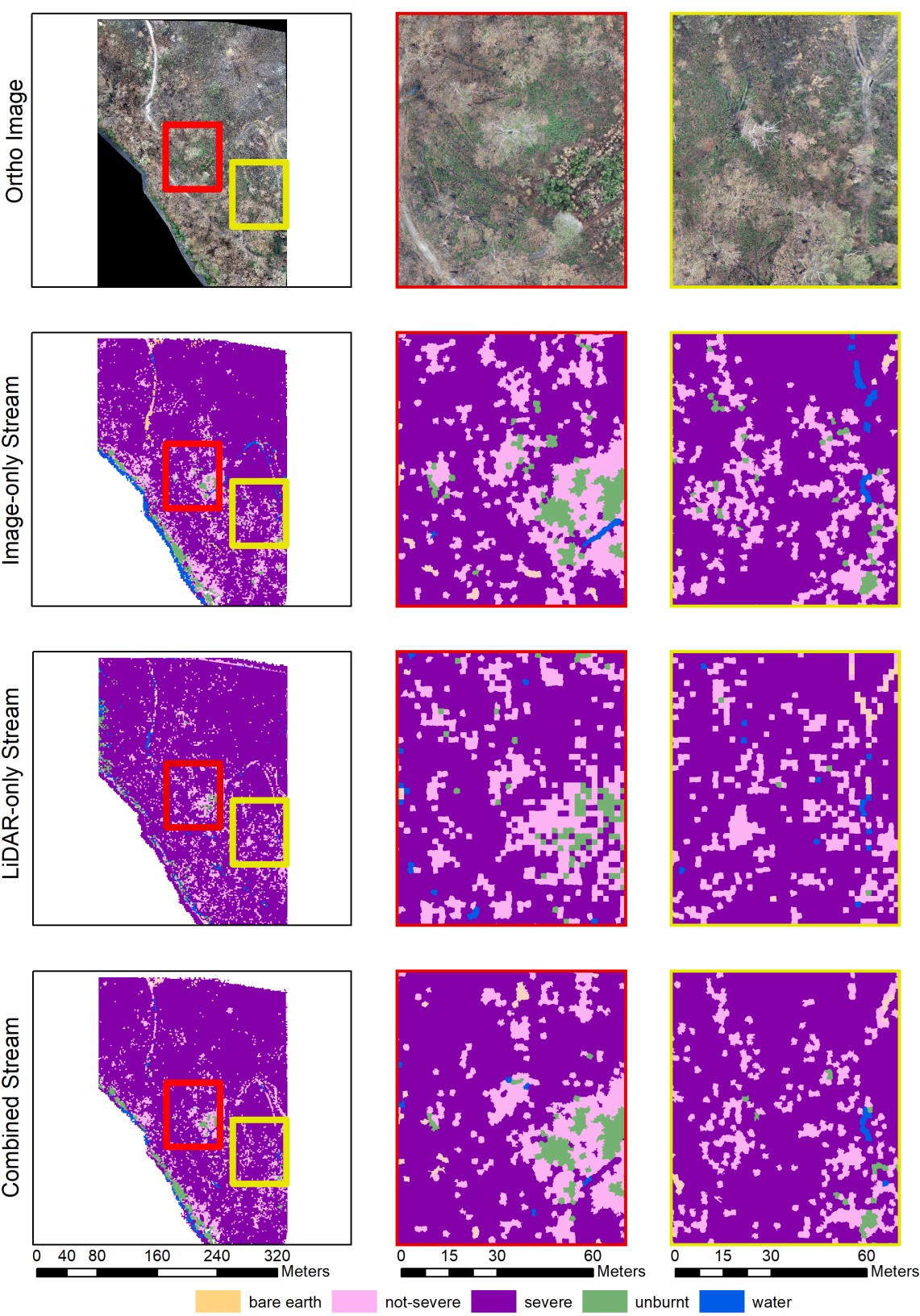

**Figure 3.** Fire severity maps produced from image-only, LiDAR-only and combined data streams.

When considering the variables selected through the feature selection process, variables derived from the texture of the canopy height model or direct structure estimates (height, layer count and volume) were used in the RF classifiers in all streams. However, different structural variables were selected across the streams. We found that in the combined and LiDAR-only streams, multi-temporal variables describing difference in CHM texture and structure metrics were selected to describe severity in sedgeland areas. This is in contrast to the image-only stream, which used only the difference in texture metrics of the CHM. The combined and image-only stream used variables that described the reflectance characteristics as well as the texture of the ortho image.

**Table 8.** Confusion matrix for image-only data stream describing severity of sedgeland segments.

| | | Reference Data—Pre and Post Variables | | | |
|---|---|---|---|---|---|
| | Class | Not-Severe | Severe | Unburnt | User's Accuracy |
| **Classified Data—Pre and Post Variables** | Not-severe | 7 | 27 | 1 | 20.0% |
| | Severe | 34 | 177 | 10 | 80.1% |
| | Unburnt | 1 | 0 | 8 | 88.9% |
| | Producer's Accuracy | 16.7% | 86.8% | 42.1% | Overall: 72.4% |

**Table 9.** Confusion matrix for LiDAR-only data stream describing severity of low vegetation segments.

| | | Reference Data—Pre and Post Variables | | | |
|---|---|---|---|---|---|
| | Class | Not-Severe | Severe | Unburnt | User's Accuracy |
| **Classified Data—Pre and Post Variables** | Not-severe | 12 | 15 | 2 | 41.4% |
| | Severe | 32 | 191 | 18 | 79.3% |
| | Unburnt | 0 | 1 | 3 | 75.0% |
| | Producer's Accuracy | 27.3% | 92.3% | 13.0% | Overall: 75.2% |

**Table 10.** Confusion matrix for Combined stream describing severity of sedgeland segments.

| | | Reference Data—Pre and Post Variables | | | |
|---|---|---|---|---|---|
| | Class | Not-Severe | Severe | Unburnt | User's Accuracy |
| **Classified Data—Pre and Post Variables** | Not-severe | 10 | 16 | 4 | 33.3% |
| | Severe | 31 | 188 | 10 | 82.1% |
| | Unburnt | 1 | 0 | 5 | 83.3% |
| | Producer's Accuracy | 23.8% | 92.2% | 26.3% | Overall: 76.6% |

### 3.2.2. Classification of Severity within Forest Segments

The accuracy of the severity classification within forest for all three data streams classified were within 4% (image-only: 76.6%; LiDAR-only: 74.5%; and combined: 78.5%) Figure 3).

Similar to the classification of severity in sedgeland segments, all data streams had high producer's and user's accuracy for segments classified as severe in comparison to those classified as not severe (Tables 11–13). Producer's and user's accuracy for severe segments were within 6% across all streams. Not-severe segment user's and producer's accuracy was highest with the LiDAR-only stream whilst unburnt producer's and user's accuracy were highest in the image-only and combined data streams.

The predictor variables used in the modelling of severity in forest areas utilised structural, texture and ortho image metrics (Appendix B). The image-only stream used

six variables whilst the LiDAR-only and combined streams used 14 predictor variables (Appendix B. When applied to the training data, the feature selection resulted in four post variables and two variables describing difference between pre- and post-fire being used from the image-only stream. In contrast the LiDAR-only stream used one variable derived from the post-fire capture, seven variables from the pre-fire capture and six variables describing the difference between pre- and post-fire captures. The combined stream used five variables from post-fire capture, two variables from pre-fire capture and six variables describing the difference between pre- and post-fire variables.

When analysing the variables selected, all streams used variables describing the volume of the point cloud either pre or post fire. Similar to the derivation of severity in sedgeland areas, the predictor variables were not consistent across the three streams. Predictor variables describing the texture of the CHM and ortho image were selected across all three streams. The LiDAR-only stream was the only stream that used variables describing the height pre-fire and relative change in height and volume between data captures. Further, metrics describing changes in the texture variables of the CHM were only used in the LiDAR-only and combined data streams.

**Table 11.** Confusion matrix for image-only stream describing severity of forest segments.

| | | Reference Data—Pre and Post Variables | | | |
|---|---|---|---|---|---|
| | Class | Not-Severe | Severe | Unburnt | User's Accuracy |
| Classified Data—Pre and po-st vari-ables | Not-severe | 12 | 22 | 5 | 30.8% |
| | Severe | 29 | 182 | 5 | 84.3% |
| | Unburnt | 1 | 0 | 9 | 90.0% |
| | Producer's Accuracy | 28.6% | 89.2% | 47.4% | Overall: 76.6% |

**Table 12.** Confusion matrix for LiDAR-only stream describing severity of forest segments.

| | | Reference Data—Pre and Post Variables | | | |
|---|---|---|---|---|---|
| | Class | Not-Severe | Severe | Unburnt | User's Accuracy |
| Classified Data—Pre and Post Variables | Not-severe | 13 | 18 | 3 | 38.2% |
| | Severe | 31 | 184 | 13 | 80.7% |
| | Unburnt | 0 | 5 | 7 | 58.3% |
| | Producer's Accuracy | 29.5% | 88.9% | 30.4% | Overall: 74.5% |

**Table 13.** Confusion matrix for combined stream describing severity of forest segments.

| | | Reference Data—Pre and Post Variables | | | |
|---|---|---|---|---|---|
| | Class | Not-Severe | Severe | Unburnt | User's Accuracy |
| Classified Data—Pre and Post Variables | Not-severe | 8 | 12 | 3 | 34.8% |
| | Severe | 33 | 192 | 8 | 82.4% |
| | Unburnt | 1 | 0 | 8 | 88.9% |
| | Producer's Accuracy | 19.0% | 94.1% | 42.1% | Overall: 78.5% |

*3.3. Change in Vertical Structure as a Mechanism for Describing Fire Severity*

Visual inspection of the point clouds showed a varying capability of each respective technology to describe the vertical profile of the vegetation pre and post fire. UAS LiDAR point clouds appear to represent the canopy and below-canopy elements most comprehen-

sively, with UAS image-based point clouds providing only partial reconstruction, especially in the post-fire capture.

### 3.3.1. Forest and Severe Fire Impact

In areas that were classified as having forest and severe fire impact, the LiDAR segments showed an increase in the mean 10th and 50th percentile height values (0.48 m, 0.39 m) Figure 4 and Table 14 from the pre-fire values. The 50th percentile height of the UAS SfM point cloud increased by 0.30 m (Figure 5). The UAS SfM point clouds showed a decrease in the 10th percentile height of 3.81 m between pre- and post-fire captures (Table 14). Inspection of the point clouds highlights an example of this variation in structural representation (Figures 4 and 5). Both the UAS SfM and LiDAR point clouds showed a decrease in the 90th percentile heights of 0.39 m and 1.44 m, respectively).

The layer counts in both the LiDAR and SfM showed a decrease in the number of layers post fire (Table 14). This difference was seen to be greatest in the SfM point clouds with a mean decrease of 1.33 layers. A decrease was also observed in the volume estimate of UAS LiDAR point clouds of 0.45 m$^3$ and a larger decrease in UAS SfM point clouds of 6.61 m$^3$.

### 3.3.2. Forest and Not Severe Fire Impact

In areas that were classified as having forest and not-severe fire affects, a decrease in the mean 90th percentile heights was seen in both the UAS SfM and LiDAR point clouds (Table 14 and Figures 4 and 5). The SfM point clouds showed a decrease in heights in the 10th and 50th percentile heights. This is in contrast to the UAS LiDAR point clouds, which increased in height in these layers. The layer count showed a mean decrease of layers in the UAS LiDAR and SfM point clouds. There was a greater loss of volume in the UAS SfM point clouds in comparison to the LiDAR point clouds.

### 3.3.3. Sedgeland and Severe Fire Impact

Within the areas classified as sedgeland, the structural change in areas of severely burnt segments was observed in both the LiDAR and SfM point clouds with a decrease in all percentile heights (Table 14). The layer counts were seen to have a mean decrease within both the UAS LiDAR and SfM point clouds. The mean volume also decreased with the largest reduction occurring in the SfM point clouds (UAS SfM: 6.09 m$^3$; and UAS LiDAR: 1.01 m$^3$).

### 3.3.4. Sedgeland and Not Severe Fire Impact

In segments that were classified as sedgeland and did not burn severely, a mean decrease in all percentile heights was observed (Table 14). The SfM point clouds showed a greater mean decrease in the percentile heights in comparison to the LiDAR point clouds, especially in the 50th and 90th percentile heights.The layer counts were seen to have a mean decrease in both the UAS SfM and UAS LiDAR. Whilst both technologies showed a decrease in volume, the UAS SfM had a greater decrease with 2.90 m$^3$.

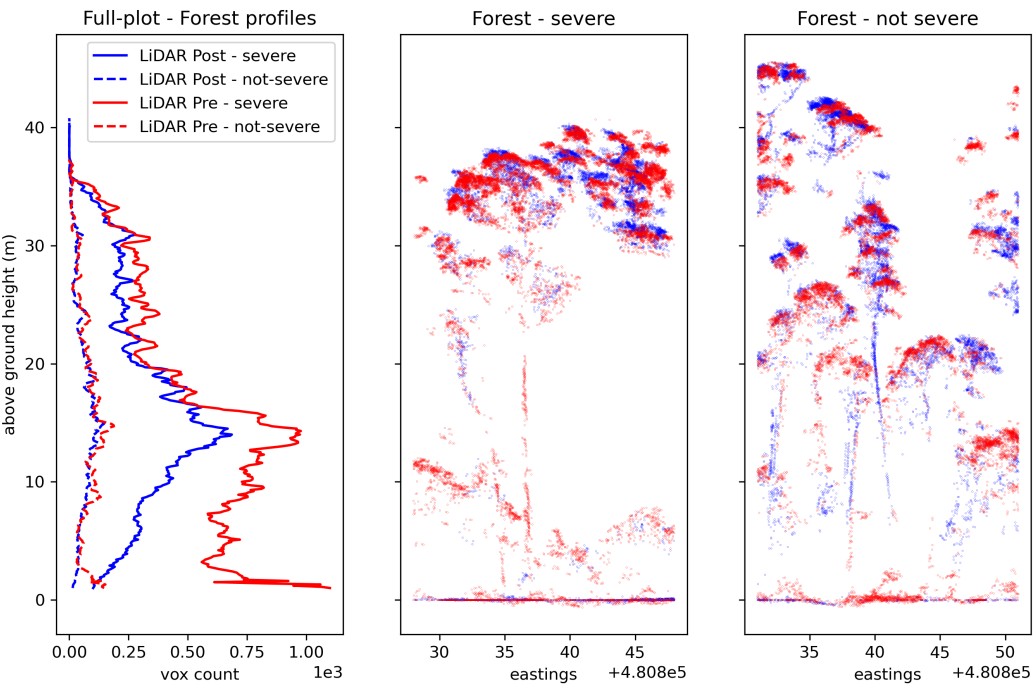

**Figure 4.** Differences in UAS LiDAR point cloud information pre- and post-fire within areas classified as forest.

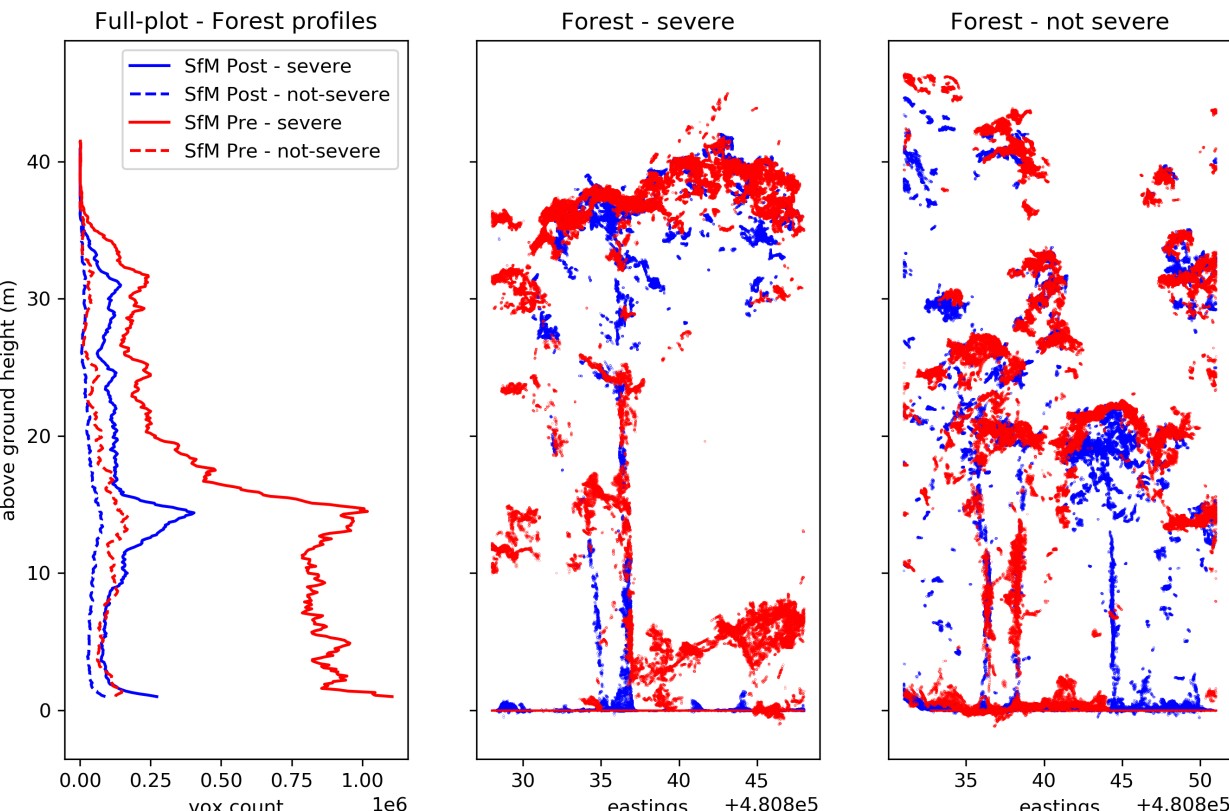

**Figure 5.** Differences in UAS SfM point cloud information pre- and post-fire within areas classified as forest.

**Table 14.** Vertical structure change separated by the classification of the vegetation and severity type.

| Capture Method | | | | LiDAR | | | | | | | SfM | | | | | | | | Difference (m) |
|---|---|---|---|---|---|---|---|---|---|---|---|---|---|---|---|---|---|---|---|
| Time | | Pre | | | | Post | | | | Difference (m) | Pre | | | | Post | | | | |
| Value | Mean | Std Dev | Skew | Kurtosis | Mean | Std Dev | Skew | Kurtosis | | Mean | Std Dev | Skew | Kurtosis | Mean | Std Dev | Skew | Kurtosis | |
| **Forest — Severe** | | | | | | | | | | | | | | | | | | | |
| 10th % height (m) | 6.57 | 7.17 | 1.87 | 3.80 | 7.05 | 8.17 | 1.04 | 0.13 | 0.48 | 5.39 | 7.34 | 2.55 | 7.05 | 1.58 | 3.90 | 4.11 | 18.52 | −3.81 |
| 50th % height (m) | 20.57 | 10.81 | −0.10 | −1.07 | 20.96 | 10.96 | −0.29 | −0.90 | 0.39 | 15.80 | 10.51 | 0.45 | −0.84 | 16.10 | 11.34 | 0.02 | −1.36 | 0.30 |
| 90th % height (m) | 27.32 | 9.57 | −0.23 | −0.80 | 26.94 | 10.73 | −0.63 | −0.09 | −0.39 | 26.16 | 10.03 | −0.51 | −0.47 | 24.72 | 11.91 | −0.62 | −0.59 | −1.44 |
| Layer Count | 4.86 | 1.90 | 0.58 | 0.96 | 4.19 | 2.18 | 0.30 | −0.04 | −0.68 | 4.49 | 1.92 | 0.28 | 0.26 | 3.16 | 2.00 | 0.49 | −0.10 | −1.33 |
| Volume (m$^3$) | 2.65 | 1.27 | 0.38 | 0.23 | 2.20 | 1.29 | 0.80 | 1.26 | −0.45 | 11.66 | 4.13 | −0.35 | 0.85 | 5.05 | 3.48 | 0.90 | 1.06 | −6.61 |
| **Forest — Not-Severe** | | | | | | | | | | | | | | | | | | | |
| 10th % height (m) | 7.02 | 6.94 | 1.54 | 2.15 | 8.99 | 8.24 | 0.59 | −0.77 | 1.97 | 5.03 | 7.28 | 2.72 | 7.50 | 2.27 | 3.80 | 3.01 | 10.21 | −2.76 |
| 50th % height (m) | 23.81 | 10.27 | −0.38 | −0.76 | 24.05 | 10.22 | −0.51 | −0.58 | 0.24 | 18.28 | 10.45 | 0.24 | −0.83 | 19.76 | 9.71 | −0.32 | −0.95 | 1.49 |
| 90th % height (m) | 30.15 | 9.64 | −0.42 | −0.61 | 30.04 | 10.22 | −0.68 | 0.03 | −0.10 | 28.89 | 10.53 | −0.69 | −0.12 | 29.23 | 10.28 | −0.78 | 0.06 | 0.34 |
| Layer Count | 4.98 | 1.91 | 0.41 | −0.22 | 4.63 | 2.07 | 0.25 | −0.09 | −0.35 | 4.79 | 1.94 | 0.06 | 0.04 | 3.93 | 1.83 | 0.18 | −0.32 | −0.86 |
| Volume (m$^3$) | 3.64 | 1.30 | 0.10 | 0.01 | 3.36 | 1.52 | 0.43 | 0.19 | −0.27 | 13.95 | 4.67 | −0.76 | 1.19 | 9.50 | 4.55 | 0.73 | 0.60 | −4.45 |
| **Sedgeland — Severe** | | | | | | | | | | | | | | | | | | | |
| 10th % height (m) | 0.48 | 0.80 | 11.19 | 249.66 | 0.23 | 1.49 | 10.86 | 125.90 | −0.25 | 0.84 | 1.05 | 13.98 | 365.31 | 0.15 | 0.89 | 15.46 | 274.95 | −0.69 |
| 50th % height (m) | 2.30 | 4.24 | 4.58 | 23.34 | 1.94 | 5.20 | 3.75 | 14.66 | −0.36 | 2.20 | 2.93 | 6.25 | 50.41 | 1.52 | 4.22 | 4.52 | 22.66 | −0.68 |
| 90th % height (m) | 5.52 | 6.96 | 2.20 | 5.25 | 4.19 | 7.41 | 2.42 | 5.77 | −1.33 | 5.62 | 6.48 | 2.38 | 6.23 | 3.72 | 6.53 | 2.68 | 7.87 | −1.90 |
| Layer Count | 1.57 | 1.53 | 1.43 | 3.99 | 0.83 | 1.20 | 2.19 | 6.41 | −0.75 | 1.84 | 1.25 | 1.35 | 2.98 | 0.74 | 1.03 | 2.30 | 9.40 | −1.10 |
| Volume ((m$^3$) | 1.64 | 1.21 | −0.17 | −1.05 | 0.64 | 0.82 | 1.53 | 4.50 | −1.00 | 7.12 | 3.28 | −0.02 | 1.03 | 1.10 | 1.63 | 2.39 | 9.40 | −6.02 |
| **Sedgeland — Not-Severe** | | | | | | | | | | | | | | | | | | | |
| 10th % height (m) | 1.08 | 1.28 | 5.42 | 70.09 | 0.54 | 1.32 | 9.01 | 115.19 | −0.54 | 1.22 | 1.16 | 3.19 | 29.90 | 0.49 | 1.43 | 12.25 | 189.11 | −0.74 |
| 50th % height (m) | 3.37 | 4.47 | 3.88 | 18.53 | 3.45 | 5.46 | 3.22 | 11.66 | 0.08 | 2.86 | 3.03 | 5.21 | 44.98 | 2.42 | 4.00 | 4.19 | 22.01 | −0.44 |
| 90th % height (m) | 7.20 | 7.51 | 1.82 | 3.31 | 6.82 | 7.75 | 1.91 | 3.73 | −0.38 | 6.74 | 6.96 | 1.99 | 4.37 | 5.21 | 6.59 | 2.38 | 6.94 | −1.53 |
| Layer Count | 1.87 | 1.68 | 1.24 | 1.79 | 1.36 | 1.26 | 1.60 | 4.05 | −0.51 | 1.84 | 1.44 | 1.10 | 1.54 | 1.07 | 1.13 | 2.07 | 9.78 | −0.77 |
| Volume (m$^3$) | 1.69 | 1.14 | −0.09 | −0.04 | 1.45 | 1.19 | 0.47 | −0.01 | −0.24 | 6.82 | 4.08 | −0.36 | −0.71 | 3.92 | 3.84 | 0.64 | 0.15 | −2.90 |

## 4. Discussion

This study presented an evaluation of UAS LiDAR and image-based point cloud derived variables using a supervised classification to produce maps of land cover and fire severity. Temporally coincident observations were captured across a range of structurally diverse vegetation communities, allowing for a direct comparison between the two data sources and processing streams. Furthermore, the area was captured both pre- and post-fire allowing for a two stage classification: firstly classifying land cover and secondly classifying the severity within each land cover type, providing a testbed to explore the changes resulting from fire. Prior work from fixed wing and satellite remote sensing platforms have demonstrated the utility of imagery and supervised classifications to estimate fire severity across an area [43,44,46,68–70,79]. McKenna et al. [24], Simpson et al. [26] and Carvajal-Ramírez et al. [27] demonstrated the utility of UAS SfM image-derived variables from pre- and post-fire point clouds to map fire severity at local scales across areas with limited structural diversity (open grassland, woodland and peatland). Similarly, Arkin et al. [25] utilised UAS SfM workflows to derive image and structural variables captured post-fire, in combination with a supervised classification, to map fire severity across a burnt forested area (Douglas fir, hybrid white spruce and lodgepole pine). This study extends this research by comparing the utility of LiDAR-only, image-only and combined data streams separately, to classify vegetation and severity in a structurally diverse study area.

### 4.1. Land Cover Accuracy

Confusion matrices showed similar (within 5%) overall, producer's and user's accuracy for the land cover classification accuracy across the three data streams of processing. Consistent with Goodbody et al. [80] and Feng et al. [81], analysis of the variables used to map land cover in each of the three data streams demonstrated that all streams utilised texture metrics to identify different land cover classes. Whilst the workflow presented here classified land cover into four categories, land cover transition zones were noted by assessors as being difficult to classify through visual assessment. More broadly, the use of imagery and point cloud data provides new opportunities to classify land cover that takes into a consideration a diverse array of factors beyond that which human interpretation is able to achieve.

Visual inspection of the UAS LiDAR and UAS image-based point clouds captured pre-fire demonstrated that both technologies were able to adequately describe the vertical profile of the vegetation (Figures 4 and 5). Whilst this reconstruction of below-canopy vegetation supports prior research that demonstrated the ability of UAS LiDAR point clouds to represent forest structure in a variety of forest types [38,39,42,82], it is in contrast to previous studies showing that UAS image-based point clouds were not able to represent information beneath the canopy accurately [40,42]. Potential reasons for increased vegetation representation beneath the canopy in our dataset were the environmental conditions at the time of capture, with low wind and good lighting beneath the canopy allowing for strong contrast between the ground and trees, assisting the point cloud reconstruction. Additionally, in the canopy areas of the plot, a greater amount of vegetation in the mid-storey/elevated layers in comparison to the post-fire capture is likely to have aided the depth reconstruction by providing extra features for the depth matching process. Prior research has demonstrated greater reliability of UAS LiDAR point clouds in generating a point cloud due to the active nature of the sensor being less sensitive to illumination conditions [36,40].

### 4.2. Severity Accuracy

In all data streams, and for both forest and sedgeland classes, classification of severe segments was more accurate than not-severe segments. This trend was also shown by McKenna et al. [24] who highlighted that the high severity classes attained a higher accuracy than low severity and unburnt classes. This potentially indicates an underlying bias in the dataset used in this analysis with the majority of the plot being severely burnt and being

more obvious to detect. Further reasons for misclassification of not-severe segments may be in the form of obscuration of fire affected layers by a taller canopy. Hyper-emergence of individual trees is common to wet forests, and in this scenario it may have limited observations of areas that have had minimal fire impact [83].

Validating remotely sensed metrics of vegetation classification and fire severity with ground observations at the point scale is considered best practice. However, it is challenging to implement over large areas and requires ecological expertise. The visual interpretation of high-resolution ortho images for the determination of severity has been shown to be strongly correlated with field-based measures of severity [84,85]. It is acknowledged that visual interpretation limits the assessment of fire severity to what is visible in the imagery and excludes variables such stem scorch and understorey loss in areas of closed canopies. Previous research utilising UAS ortho imagery for the determination of fire severity have utilised visual interpretation as a reference for classification accuracy [24,25]. To further ensure that a high level of precision was obtained in the severity assessment in this manuscript, at least two assessors must have the same severity assessment. The classification of severity using broad user-defined scales potentially limits the degree to which fire severity can be classified. Further work could investigate the ability of UAS-derived variables and machine learning processes to deal with multiple classes such as those by Collins et al. [69] and Tran et al. [86]. However, we acknowledge that there is a likely trade-off that exists between the number of categories and the ability for interpreters to accurately distinguish between these categories. The timing of the post-fire capture is also important to consider in the context of severity accuracy. Post-fire rainfall at the study area led to a flush of growth, which is likely to have decreased the accuracy of the classification, with areas assessed as high severity confounded with spectral characteristics similar to pre-fire vegetation.

Predictor variables derived from point clouds were used in all streams for mapping fire severity either directly from percentile heights, layer count and volume estimates or indirectly through the production of canopy height models. Analysis of the predictor variables used in each classifier demonstrates that there was no consistent set of structural predictor variables used across all streams. Variables describing differences in texture between pre- and post-fire were selected by mapping severity across the plot in all streams. It was hypothesised that the improved vegetation representation from LiDAR would mean that predictor variables describing height or layer count differences between pre-fire and post-fire would be used in the prediction of severity, particularly in areas of forest. This would support Hu et al. [43], Hoe et al. [44] and Skowronski et al. [46], who demonstrated the effectiveness of describing changes to structural characteristics such as profile area and LiDAR return proportions 2 m above ground, pre-fire 95% heights and pre-fire return proportions 2 m above ground. However, the structural variables generated in this research showed only a small change between pre- and post-fire (Table 14). This may indicate that the variables that are commonly used to assess structure at not suitable for fire induced impact assessments in the forest types observed in this study unless there is full tree loss. Whilst large amounts of fine fuel are consumed during a fire, the Eucalypt forests surveyed in this study have structure that persists after fire [87,88]. For the metrics utilised in this research to be selected through the feature selection process, it is predicted that more significant structural change is needed such as tree fall as is observed in some North American forests [89].

The models in each stream estimating severity in forest and sedgeland areas utilise variables derived from both pre- and post-fire captures. The image-only and combined streams used a greater number of post-fire factors in comparison to the LiDAR-only stream when predicting severity in forest areas. Variables that describe the difference between the captures were also utilised which supports prior work highlighting the effectiveness of bi-temporal observations to assess severity [24,44,84,90]. Further work should investigate the relative contribution of pre- and post-fire predictor variables in estimating fire severity. Recent work by Hoe et al. [44] and Skowronski et al. [46] explicitly links pre-fire fuel

loading with fire severity, representing an opportunity to improve potential fire predictions across landscapes when combined with modelled weather conditions. This further work should also consider the findings of Arkin et al. [25], who used only post-fire variables in the mapping of fire severity which would enhance the usability of the workflow where pre-fire data is unavailable.

The overall classification accuracy of severity in forest areas (Image only: 76.6%; LiDAR only 74.5%; and Combined 78.5%) and sedgeland areas (Image only: 72.4%; LiDAR only: 75.2%; and Combined: 76.6%) in this study was achieved with very high-resolution (0.02 m) data. Comparatively, satellite derived assessments of fire severity are completed at regional scales where pixel values describe areas between 3 and 500 m [21,91,92]. Previous studies have demonstrated high-resolution satellite imagery is capable of severity classification accuracy within the range of 50% and 95% [21]. Similar accuracy is achievable from imagery captured from manned aircraft, however, this data runs into similar issues in capturing understorey change especially in dense canopy environments [85,93]. Point cloud information derived from UAS SfM workflows has been shown to provide information describing changes in understory in open canopy forests present in this study and previous work [24].

LiDAR captured from manned aircraft go someway to address this issue with greater capacity to describe changes in below canopy vegetation structure with severity classification accuracy shown to be between 51% and 54.9% in mixed-conifer, oak woodlands and hardwood-evergreen forests [44]. Point cloud information can be used to detect changes in structure from fire at the tree level and at sub-tree scale and shrub level Figure 4 [42]. Additionally, UAS may be flown at the time desired by the operators. This is particularly useful in situations where there may be the opportunity to collect information prior to the passing of a fire and/or in diverse or transitional ecosystems, where the post-fire vegetation condition must be captured within a few days (e.g., grasslands and tropical savannas, 5–6 days) [94] or weeks (e.g., dry sclerophyll forests of southern Australia) [95] to enable severity to be accurately characterised. In these scenarios, high-spatial and temporal resolution products derived from UAS may be particularly useful to validate low-resolution, yet wide area satellite or airborne derived products [96].

*4.3. Vertical Profile*

As described by Hillman et al. [40] and Wallace et al. [36], visual inspection of the pre- and post-fire UAS LiDAR point clouds provide a complete representation of the vertical profile and allow for a description of forest structure in all strata. Similarly, the pre-fire UAS SfM point clouds appeared to provide a complete representation of the vertical profile. This is in contrast to the UAS SfM point clouds derived from the post-fire capture which provide limited reconstruction of the vegetation. This is demonstrated by large decreases in the mean height of the 50th and 90th percentile heights. The lack of information content in the post-fire UAS SfM point clouds could be due to similar factors as observed by Hillman et al. [40], with poor contrast between burnt ground and vegetation, and wind conditions at time of capture. These factors can confound the image matching process, resulting in limited vegetation reconstruction and have the potential to inaccurately represent structural change.

In contrast, the UAS LiDAR point clouds derived from the post-fire capture showed an increase in the 10th and 50th percentile heights. Whilst vegetation heights are expected not to have increased post-fire, the increase in these percentile heights is likely to be due to an increased penetration of the sensor beam through a sparser canopy. Whilst the differences in LiDAR sensors used between the pre- and post-fire data capture campaigns may contribute to small discrepancies in the height estimations, this is not believed to be a consideration that would influence the accuracy, as only the first returns from each sensor were used. This highlights an opportunity for further work to consider the use of all returns when deriving structural measurements with the potential that more information may be yielded.

Despite greater information content present in the UAS LiDAR point clouds, the feature selection process utilised fewer direct structural variables in the final mapping of land cover and severity. Further work should look to develop metrics that maximise the different information content contained within the UAS LiDAR point clouds. One such area that may yield new insights is the characterisation of ladder fuels and vertical connectivity. Wilkes et al. [67], for example, derived the number of layers in each segment and provided an indication of the presence and absence of vegetation in the point cloud. Approaches for deriving metrics that describe the vegetation and/or fuel properties over the vertical profile could be used to quantify the presence, change and consumption of ladder fuels. Approaches to quantify structure and arrangement in the vertical profile in previous studies have typically combined qualitative and quantitative approaches to measuring fuels [97–99], with some preliminary studies utilising remote sensing to measure canopy base height, percentage cover below canopy or fuel gaps [100–104]. Fuel strata gap, as proposed by Cruz et al. [105], is one such method that could be applied to leverage the available information content. However this method, whilst effective in North American forest types, may not be as successful in Eucalypt forest types where the arrangement of fuel is multi-layered and complex. Similar to the work presented by Skowronski et al. [104] and the approach implemented by Hillman et al. [40], this may allow for the identification and quantitative representation of ladder fuels independent of forest type.

### 4.4. Operational Applicability

UAS are being increasingly used in forest and fire management to measure landscape condition and for real-time emergency observations [106–111]. The versatility of UAS is that they can be deployed quickly and efficiently post-fire to collect severity information. Careful consideration of the purpose of the assessment should be made so that the sensor payload matches the desired information outputs. For example, this research demonstrated that UAS SfM point clouds cannot be relied upon to represent structural change from fire. Conversely, UAS LiDAR point clouds provided a more complete representation of vegetation structure pre- and post-fire. Whilst both technologies had difficulty in discerning not-severe areas from severe, the high accuracy in the severe category alone allows land managers to identify priority areas of treatment, without the need for costly airborne image capture.

The capacity to accurately map fire severity will enhance land managers' understanding of ecosystem response. Given the reliability of detecting below-canopy vegetation structure in UAS LiDAR point clouds this technology provides the greatest opportunity to measure post-fire vegetation traits in the complex wet-eucalypt forest ecosystems. Utilising high-resolution measurements from UAS LiDAR facilitates the precise estimation of foliar change from fire. When high-resolution UAS-derived estimates of fire severity are considered as part of an ensemble approach to measuring fire severity from satellite, fixed-wing, ground-based and remotely-piloted platforms, these inputs can then be used to train models of severity and hazard over much larger areas such as those presented in [69,112,113]. When combined with pre-fire fuel hazard information, UAS LiDAR point clouds may allow us to untangle the effect of fuel hazard and structure on flammability and fire severity, which is poorly understood in wet forest systems [114–116]. High-resolution fire severity assessments can also be used to evaluate and inform treatment practices (e.g., prescribed fire and timber harvesting) [116–119]. With an accurate understanding of how comprehensively the vegetation has been affected by fire, development of more accurate fuel accumulation curves are also able to be developed, which is critical for future fire management. Additionally, as fire behaviour modelling is enhanced through the use of physics-based approaches, accurate 3D vegetation descriptions of on-ground fuel properties will allow fire managers to generate more accurate fire behaviour simulations, effectively deploy first responders and implement fuel management practices [120–123].

## 5. Conclusions

With an increasing frequency and severity of fires, there is a growing need to understand the severity of fire and associated recovery of vegetation post-fire. To the authors' knowledge, there have been no prior studies utilising UAS LiDAR-derived variables with supervised classification to map land cover type and fire severity. This research contributes to this gap in knowledge and demonstrates the utility of metrics derived from UAS LiDAR point clouds captured pre- and post-fire to map vegetation and severity. Through a feature selection process, we selected subsets of predictor variables to build classifiers that used a small number of variables for the classification of land cover and fire severity. A comparison was made to image-only and combined (UAS LiDAR and UAS image predictor values) data streams with UAS LiDAR derived variables. The results indicate that UAS LiDAR provided similar overall accuracy to UAS image and combined data streams to classify severity in areas of forest with canopy dominance (UAS image: 76.6%; UAS LiDAR: 74.5%; and Combined: 78.5%) and areas of sedgeland (UAS image: 72.4%; UAS LiDAR: 75.2%; and Combined: 76.6%). Analysis of structural variables in combination with visual inspection of point clouds derived from image-based and LiDAR point clouds highlighted a greater level of vegetation reconstruction in the LiDAR point clouds. This observation is significant for mapping fire severity. Despite the feature selection process and subsequent accuracy analysis highlighting the similar capacity of each technology to classify fire severity, large differences in the information content indicate that the metrics derived for describing structural change in this study area were not suitable to represent the consumption of fine fuel. Future work should investigate the capacity of UAS-derived products to represent fine-fuel and develop metrics that are able to represent this change of vegetation beneath the canopy. The analysis presented in this paper demonstrates the capacity of UAS LiDAR point clouds to map land cover and severity from which land managers can make key decisions for identifying high priority areas post fire.

**Author Contributions:** Conceptualisation, S.H., B.H. and L.W.; Data curation, S.H., B.H., L.W., D.T., A.L. and K.R.; Formal analysis, S.H., B.H. and L.W.; Funding acquisition, K.R. and S.J.; Investigation, S.H., B.H., L.W., D.T. and A.L.; Methodology, S.H., B.H., L.W. and A.L.; Project administration, L.W. and K.R.; Resources, D.T., A.L. and S.J.; Software, S.H., B.H. and L.W.; Supervision, L.W., K.R. and S.J.; Validation, S.H., B.H. and L.W.; Visualisation, S.H., B.H. and L.W.; Writing—original draft, S.H.; and Writing—review and editing, S.H., B.H., L.W., D.T., A.L., K.R. and S.J. All authors have read and agreed to the published version of the manuscript.

**Funding:** This research was funded by the Bushfire Natural Hazard CRC (CON/2017/01377).

**Institutional Review Board Statement:** Not applicable.

**Informed Consent Statement:** Not applicable.

**Data Availability Statement:** The data presented in this study are available on request from the corresponding author. The data are not publicly available due to ongoing research and development using these datasets.

**Acknowledgments:** The support of the Commonwealth of Australia through the Bushfire and Natural Hazards Cooperative Research Centre and the Australian Postgraduate Award is acknowledged. The University of Tasmania and TerraLuma research group are gratefully acknowledged for providing their equipment, lab and expertise.

**Conflicts of Interest:** The authors declare no conflict of interest.

## Appendix A. Predictor Variables Used in Land Cover Calculation

**Table A1.** Summary of accuracy derived from the test and validation datasets for each of the three streams of data and predictor variables used in each stream.

|  | Image-Only | LiDAR-Only | Combined |
|---|---|---|---|
| Validation | 80.6% | 78.9% | 83.1% |
| Variables used | 90th percentile height<br>Distance between top 2 layers (SfM)<br>A (Green-red) mean (Ortho)<br>B (Blue-yellow) mean (Ortho)<br>Homogeneity (CHM-SfM)<br>Entropy (CHM-SfM)<br>Contrast (Ortho)<br>Correlation (Ortho) | Layer count<br>10th percentile height<br>50th percentile height<br>Correlation (CHM)<br>Homogeneity (CHM) | 50th percentile height (SfM)<br>10th percentile height (LiDAR)<br>90th percentile height (LiDAR)<br>Distance between top 2 layers (LiDAR)<br>A (Green-red) mean (Ortho)<br>B (Blue-yellow) mean (Ortho)<br>Sum of squares variance (CHM-SfM)<br>Homogeneity (CHM-SfM)<br>Contrast (Ortho)<br>Correlation (Ortho) |

**Table A2.** Summary of accuracy derived from the validation dataset and predictor variables used in Image-only stream.

| Stream | Severity | |
|---|---|---|
| **Image Stream** | **Forest** | **Sedgeland** |
| Validation | 75.8% | 72.8% |
| Variables used | Volume (Post)<br>A (Green-red) mean (Post)<br>B (Blue-yellow) mean (Post)<br>Correlation (CHM-Post)<br>Correlation difference (CHM)<br>A (Green-red) mean difference (Ortho) | Volume (Post)<br>10th percentile height (Post)<br>A (Green-red) mean (Post)<br>B (Blue-yellow) mean (Post)<br>A (Green-red) mean (pre)<br>Correlation (CHM-Post)<br>Sum of squares variance (CHM-Post)<br>Correlation (Ortho-Post)<br>Homogeneity (Ortho-Post)<br>Contrast difference (CHM)<br>Homogeneity difference (CHM)<br>Homogeneity difference (Ortho)<br>A (Green-red) mean difference<br>B (Blue-yellow) mean difference |

## Appendix B. Predictor Variables Used in Severity Classification from Pre and Post-Fire Calculation

**Table A3.** Summary of accuracy derived from the validation dataset and predictor variables used in LiDAR-only stream.

| Stream | Severity | |
|---|---|---|
| **LiDAR Stream** | **Forest** | **Sedgeland** |
| Validation | 74.5% | 75.2% |
| Variables used | Volume (Pre)<br>10th percentile height (Pre)<br>50th percentile height (Pre)<br>Entropy (CHM-Post)<br>Contrast (CHM-Pre)<br>Correlation (CHM-Pre)<br>Sum of squares variance (CHM-Pre)<br>Homogeneity (CHM-Pre)<br>Volume difference<br>10th percentile difference<br>Angular second moment difference (CHM)<br>Contrast difference (CHM)<br>Correlation difference (CHM)<br>Sum of squares variance difference (CHM) | 10th percentile height (Post)<br>Volume (Pre)<br>10th percentile height (Pre)<br>90th percentile height (Pre)<br>Contrast (CHM-Post)<br>Entropy (CHM-Post)<br>Contrast (CHM-Pre)<br>Correlation (CHM-Pre)<br>Sum of squares variance (CHM-Pre)<br>Volume difference<br>10th percentile difference<br>50th percentile difference<br>Angular second moment difference (CHM)<br>Contrast difference (CHM)<br>Correlation difference (CHM)<br>Sum of squares variance difference (CHM) |

**Table A4.** Summary of accuracy derived from the validation dataset and predictor variables used in Combined streams.

| Stream | | Severity | |
|---|---|---|---|
| **Combined Stream** | | **Forest** | **Sedgeland** |
| | Validation | 78.5% | 76.6% |
| | Variables used | Volume (SfM-Post)<br>A (green-red) mean (Post)<br>B (blue-yellow) mean (Post)<br>B (blue-yellow) mean (Pre)<br>Correlation (CHM-Post)<br>Correlation (Ortho-Post)<br>Homogeneity (Ortho-Post)<br>Contrast (Ortho-Pre)<br>Angular second moment difference (SfM-CHM)<br>Correlation difference (SfM-CHM)<br>Angular second moment difference (LiDAR-CHM)<br>Correlation difference (LiDAR-CHM)<br>Contrast difference (Ortho)<br>A (green-red) mean difference | Volume (LiDAR-Post)<br>Volume (SfM-Post)<br>A (green-red) mean (Post)<br>B (blue-yellow) mean (Post)<br>A (green-red) mean (Pre)<br>Correlation (LiDAR-CHM-Post)<br>Homogeneity (LiDAR-CHM-Post)<br>Sum of squares variance (LiDAR CHM-Pre)<br>Sum of squares variance (LiDAR CHM-Post)<br>Homogeneity (SfM CHM-Post)<br>Correlation (SfM CHM-Pre)<br>Homogeneity (SfM CHM-Pre)<br>Correlation (Ortho-Post)<br>Homogeneity (Ortho-Post)<br>Volume difference (LiDAR)<br>50th percentile height difference (LiDAR)<br>Angular Second Moment difference (CHM-SfM)<br>Contrast difference (CHM-SfM)<br>Contrast difference (CHM-LIDAR<br>Homogeneity difference (CHM-LiDAR)<br>Angular second moment difference (Ortho)<br>Homogeneity difference (Ortho)<br>A (green-red) mean difference<br>B (blue-yellow) mean difference |

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
