# Peer review of "High-Resolution Estimates of Fire Severity—An Evaluation of UAS Image and LiDAR Mapping Approaches on a Sedgeland Forest Boundary in Tasmania, Australia"

_fire, doi:10.3390/fire4010014_

Round 1
Reviewer 1 Report
This manuscript is well structured and provides a potential pathway for future real-world impact solutions to better wildfire model spreads. This can benefit many stake holders such as academia, first responders, insurance industry etc.
While I see a tremendous potential in exploring machine learning and machine vision algorithms to advance the wildfire science, there are many unanswered questions here that I would like to know the opinion of the authors about that.
General questions:
- How do you foresee the application of this type of methods in first responder’s activities? First responders’ deployments are often guided by the results of fire spread models. This requires a lot of pre fire data which needs to be analyzed. I suggest the authors include their vision about these types of real-worlds impact in the manuscript .
Specific questions about the manuscript:
- In the abstract it is mentioned “A workflow which derives novel metrics to describe vegetation structure and fire severity is developed”. I would have a friendly suggestion here to let the readers decide about the novelty of this approach. While this might be novel in this application, this type of classification techniques has widely been used in many fields of engineering.
- I think the terminology in Table 1 should change as water cannot be a vegetation class.
- The impact levels in Table 2 is confusing to me. Severe impact is associated with tall vegetation with larger than 50% crown scorch. 50% of what? their height? Their normal projected area? Or other things?
- How sensitive is your model to the prediction class? If you change 50 to 45%, do you see a significant change in the model accuracy?
- What features did you use to classify water? Extracting water features is extremely challenging and is actually a state-of-the-art research in water pollution science. I have seen RGB thresholding to be a more reliable technique for water.
- In section 2.5.1 you mentioned that a K-means crusting algorithm was used for segmentation and labeling your data. In section 2.5.4, you mention you held 30% of the data for test the trained model. To the best of my knowledge, you cannot do both!
- I am wondering why you have not used a more robust technique such as Principal Component analysis to find the best predictors instead of removing the correlated values in section 2.5.4.
- This one is a question for me in my own work as well. I am looking at a solution about this problem. Looking at your classifier, the data set is extremely biased toward Tall and Short vegetations. Without any training, one could create a “model” like this: Tall Veg. Predictor =128/159=80%
This model can successfully predict with 80% accuracy regardless of the input data. The value of a complicated model is marginal in my opinion.
- Point number 8 becomes more concerning as we look at other classes. A simple fraction can predict much better compared with a trained model when it comes to severity impact (92% vs 79%). This problem can be seen in most of the confusion matrix. You mention this point in your discussion, but the problem is still there. I believe there should be practical visionary suggestion/implication to improve the science.
Author Response
We thank the reviewers for their comments, we appreciate the suggestions and believe they have improved the manuscript.
Please see the attachment in response to reviewer comments.

Reviewer 2 Report
It may be helpful to consider the importance of fire severity and its importance for the renewability and sustainability of ecosystems a little more in the introduction part of the article.
The discussion section in the text should be started with numbers 4 , not 5.
In the discussion part, it will be useful to emphasize more the practical benefits of the methods used in the study.
Author Response
We thank reviewer 2 for their comments.
Please see the attachment for response to comments.

Reviewer 3 Report
In this study, Hillman et al. describe the applicability of UAS LiDAR and imagery for assessing vegetation type and fire severity using pre-fire and post-fire LiDAR data. Their results indicate that UAS LiDAR and imagery can be very useful for mapping both vegetation and fire severity. I do not have much technical expertise in the use of LiDAR, but the authors appear to go into a good amount of detail in the methods and results in describing the effectiveness of UAS LiDAR, and I can see the incredible value in this new technology in assessing fire severity. Furthermore this study provides a great example of how UAS LiDAR can be effective in complex vegetation types.
However, there are currently two major shortcomings I’ve identified in this paper:
Firstly, this paper lacks a comprehensive contextualisation. Why is understanding fire severity important, both from an ecological and management perspective? The authors essentially define fire severity in the introduction then move straight into discussing how to measure it without any discussion of why fire severity matters. This is important as it sets the stage for why these methods are so valuable. I suggest adding a paragraph describing why fire severity matters. Also the discussion of operational applicability should discuss explicitly how better measurements of fire severity will improve management (see the specific comment below for details). Attiwill (1994); Bennett et al. (2016); Bowman et al. (2013); Poulos, Barton, Slingsby, and Bowman (2018); Taylor, McCarthy, and Lindenmayer (2014) are all good starting points. Further there is not much contextualisation in terms of how UAS LiDAR is an improvement from other forms of remote sensing. You mention that the resolution is much better in UAS LiDAR, can you quantify this? How much better is the resolution. More importantly you provide a bunch of accuracy metrics to quantify the ability of UAS LiDAR to assess fire severity and vegetation, but it’s hard to know how accurate these methods actually are without some form of contextualisation. How do these accuracy metrics compare to those of Airborne LiDAR or satellite based severity or vegetation assessments? What does this comparison say about the advantages of UAS LiDAR? This would be very useful to managers or other scientists who frequently use remotely sensed data are not intimately familiar with all the details of how it is produced. This should be included in the Discussion.
Secondly, there is a concerning lack of understanding of the ecosystem which is being measured. In describing the vegetation in this study, two of the three species the authors list don’t even occur in Tasmania, and they fail to even mention the most important dominant species in these systems (other than buttongrass). Further, there is no acknowledgement of the ecological importance of these vegetation. The transitional zone between buttongrass and wet forests has formed the basis alternative stable state theory, and hence is extremely important to ecological study (Bowman & Perry, 2017; Jackson, 1968). There are many other aspects of this ecosystem that make it particularly interesting, including its structural complexity (which the authors only briefly touch on), and the varying flammabilities of its vegetation. Describing these aspects of the ecosystems being studied will, in my mind, only enhance this paper, especially in terms of explaining why assessing the capabilities of UAS LiDAR is so important in the context of this specific study.
Lastly, there is a lot of jargon in this paper. I recognise a big portion of the target audience will be folks with remote sensing expertise, for whom all of this may be easily understandable. However Fire is not a remote sensing journal and I think it would make the paper a lot more accessible to, at least in the Introduction and Discussion, explain what the more niche terms mean, or a least what they are measuring. I’ve identified a few of these in the specific comments but I recommend the authors go through and check for other jargon.
The study clearly has lots of value for the fire science community, and it showcases an exciting new method for assessing fire severity. However in it’s current form, I don’t think this paper will be of much interest to anybody who does not work in the remote sensing space, given the lack of contextulisation and discussion of the ecological and management implications. Therefore I cannot recommend publication of this paper without major changes to the manuscript.
Specific Points:
L 11 I think you may struggle to find E. nitens in Tasmania. The only E. nitens you will find in Tasmania is in plantations. The authors are probably thinking of E. nitida. Even then E. nitida rarely grows taller than 20m. The tall forests in this area are dominated by E. obliqua, E. globulus, and E. regnans. The nitida dominates the transitional area between buttongrass and tall wet forest.
L. 29 This is a pretty narrow definition of fire severity. Generally severity is more broadly described as the effect of fire on vegetation, one aspect of this can be consumption of biomass, but one can also consider things like mortality and scorch.
L. 32 It’s not entirely clear which of the methods you describe after this are the direct methods and which are the indirect ones.
L. 32 This sentence and the next are confusing, how are the strata related to the visual assessments, and it’s not clear from the way you word it that the surface fuels are one of the strata. This needs clarification Maybe name all four strata and describe the qualitative assessments that can be done for each?
L. 50 Should be ‘independently’ not ‘independent’
LL. 57-61 You use a whole bunch of terminology here related to indices and ratios here, a lot of these are beyond level of expertise (I haven’t heard of most of them) but it might make this paper more accessible to a wider audience (esp. managers and ecologists) if you can preface this with a brief description of what they are measuring
L. 71 What does manual defoliation mean?
L. 73 I don’t quite understand what this sentence means. Do you mean the usefulness of remotely sensed multi-temporal structural data in assessing fire severity? I don’t think you can say you’re ‘predicting’ fire severity if you are using post-fire vegetation data. I may just not be familiar the correct terminology for LiDAR studies but I feel this sentence is overly complicated and confusing.
L. 76 authors should be authors’
L. 78 Once again I think you should use ‘estimate’ or ‘assess’ rather than predict.
L. 79 Once again you introduce a fairly technical term without any description. Maybe ‘supervised classification’ is standard terminology in the remote-sensing world, but a lot of readers without that background will not know what it means
L. 89 The authors should be sure to ground-truth the vegetation in this study area. Melaleuca alternifolia is restricted to Northern NSW and Southeast QLD. As mentioned above, E. nitens are restricted to NSW and VIC. E. nitida, which is present at the study site, is rarely taller than 20m and mostly dominates the intermediate zone, intergrading with Melaleuca. Explicitly naming the vegetation type would be useful (transitional zone between buttongrass moorlands and tall wet forests). The tall forests at this site are dominated by E. obliqua and E. globulus. Might also be worth noting the dominant understorey species (e.g. Monotoca glauca, Pomederris apetela) as this is quite important in the fire ecology of wet forests.
L. 155 10m x 10m squares or 10m2 squares? This is unclear to me.
L. 156 I find this sentence quite confusing. Firstly was this done in the field? I imagine so but this needs to be explicitly stated (once again I don’t know much about standard terminology or methodology for LiDAR studies but one could read this and assume there is no field validation going on). Also it is a little unclear what you mean when you say the points were split into two collections, maybe a map of the points would clarify things. Also, how many points were there in total?
LL. 422-431 To me this is a key knowledge gap that this study is filling, yet this point has not been made in the Introduction. I would move this to the Introduction and make sure you underscore that the groundbreaking nature of this study is that you are assessing the ability of UAS LiDAR to measure fire severity in one of the world’s most structurally complex ecosystems (see Tng, Williamson, Jordan, & Bowman, 2012; Wardell-Johnson, Neldner, & Balmer, 2017)
L. 450 Could another reason be the hyper-emergent nature of wet eucalypt forest canopies and their general sparsity (see citations above)
L. 461 Could one reason be that in lower-severity areas more vegetation remains in the emergent and intermediate canopies, thus obscuring more fine-scale variations in severity lower in the forest profile?
L. 462 Process and site-specific characteristics like what? Please elaborate.
L. 473. How does the accuracy of your severity mapping for UAS LiDAR compare to that of Aerial LiDAR or satellite-derived severity metrics? ~75% accuracy seems very good to me but I have no context.
L. 487 ‘i n’ should be ‘in’
L. 497 do you mean when only post-fire data are available?
L. 542 This, as far as I can tell, is the first time any operational or ecological applicability of UAS LiDAR. That being said, this section doesn’t really connect the actual results of this study to any management implications. Why do the results of this study matter from a management perspective? If you can measure fire severity more accurately what types of treatments will this allow managers to implement, and what can it tell them about previous treatments? I recommend digging a little deeper into the link between how understanding fire severity can influence management practices. Some good starting points are Attiwill (1994); Price and Bradstock (2012); Taylor et al. (2014)
L. 554 This is currently your only connection between fire severity and ecology in this paper, as far as I can tell, and the paper you referenced is still a review about remote sensing, and doesn’t mention anything about eucalypt ecology. This would be a good place to include literature linking fire severity and forest ecology and management.
L. 558 Excellent point. Perhaps most importantly good post-fire severity measurements will allow for validation of these fire behaviour models
Papers Referenced:
Attiwill, P. M. (1994). Ecological disturbance and the conservative management of eucalypt forests in Australia. Forest Ecology and Management, 63(2-3), 301-346. doi:https://doi.org/10.1016/0378-1127(94)90115-5
Bennett, L. T., Bruce, M. J., MacHunter, J., Kohout, M., Tanase, M. A., & Aponte, C. (2016). Mortality and recruitment of fire-tolerant eucalypts as influenced by wildfire severity and recent prescribed fire. Forest Ecology and Management, 380, 107-117. doi:https://doi.org/10.1016/j.foreco.2016.08.047
Bowman, D. M., Murphy, B. P., Boer, M. M., Bradstock, R. A., Cary, G. J., Cochrane, M. A., . . . Williams, R. J. (2013). Forest fire management, climate change, and the risk of catastrophic carbon losses. Frontiers in Ecology and the Environment, 11(2), 66-67. doi:https://doi.org/10.1890/13.WB.005
Bowman, D. M. J. S., & Perry, G. L. W. (2017). Soil or fire: what causes treeless sedgelands in Tasmanian wet forests? Plant and Soil, 420(1), 1-18. doi:10.1007/s11104-017-3386-7
Jackson, W. (1968). Fire, air, water and earth–an elemental ecology of Tasmania. Paper presented at the Proceedings of the ecological society of Australia.
Poulos, H., Barton, A., Slingsby, J., & Bowman, D. (2018). Do mixed fire regimes shape plant flammability and post-fire recovery strategies? Fire, 1(3), 39. doi:https://doi.org/10.3390/fire1030039
Price, O. F., & Bradstock, R. A. (2012). The efficacy of fuel treatment in mitigating property loss during wildfires: Insights from analysis of the severity of the catastrophic fires in 2009 in Victoria, Australia. Journal of environmental management, 113, 146-157. doi:https://doi.org/10.1016/j.jenvman.2012.08.041
Taylor, C., McCarthy, M. A., & Lindenmayer, D. B. (2014). Nonlinear Effects of Stand Age on Fire Severity. Conservation Letters, 7(4), 355-370. doi:https://doi.org/10.1111/conl.12122
Tng, D., Williamson, G., Jordan, G., & Bowman, D. (2012). Giant eucalypts–globally unique fire‐adapted rain‐forest trees? New Phytologist, 196(4), 1001-1014. doi:https://doi.org/10.1111/j.1469-8137.2012.04359.x
Wardell-Johnson, G., Neldner, J., & Balmer, J. (2017). Wet Sclerophyll forests. In D. Keith (Ed.), Australian Vegetation (pp. 281-313). Cambridge, UK: Cambridge University Press.
Author Response
We thank the reviewer for their comments. We appreciate the opportunity to revise the manuscript and believe the changes have enhanced the readability and appeal to the broader Fire audience.
Please see the attachment

Reviewer 4 Report
This is a very good manuscript with just one major flaw - it is too verbose. It is very long - which in itself is not an issue - and could be cut back to be more impactful. Too many words make it so it is hard to read. The study is very good, the science sound. The presentation needs work. Here I have some suggestions.
Introduction: Shorten - the first paragraph can be cut out.
Top of page 3 - first full sentence: "To date ..." - this is a risky statement, since the field is very fast moving. I found two papers that do some of what you have done in 2019 and 2020, but they are not exactly the same. I suggest dropping this here, and stating it as you have in the Conclusion - that "to the authors knowledge ..." Despite this, it is a novel and compelling study, so this is not needed.
In general this is a good use of tables and figures. However, there is still a lot of numerical results written in the text that can either be added to a table or dropped. The paper is chock-full of results, so it may be wise to try and slim down to the required results.
Results: The first paragraph states the same thing in three sentences - 3 different ways, and contradictory.
- Sentence 1: states that veg class proportions vary
- Sentence 2: states the amount one of the classes varies
- Sentence 3: states the numbers that lead to the statements in the previous 2 sentences
- Sentence 4: states the amount the other class varies (these two can be combined, since the amount is the same)
- Sentence 5: states that there is no significant differences. This is the important result.
In the second paragraph of results: A good example of a "throw away sentence" is: "The confusion matrices highlight ..." The authors should read through the paper and find these "extra" sentences, as they just make more words that are not helpful.
At the bottom of page 11 the word "utilized" is used several times in the last two sentences. Replace with "used" to make it less redundant. ("use" is a better word to use than "utilize" when possible) (Again in last full paragraph on page 23)
Bottom of page 13: "in areas without a tall canopy (i.e. short vegetation ..." - why not just "In areas of short vegetation ..."?
After such a long set of results, I was disappointed in such a short and "sparse" conclusion section. I expected a good overview of what the salient results were. Also, the "Results" section should be labeled "Results and Discussion", or, the results should be presented in figures and tables and then the words you have in this large section called "Discussions". That may make this a better paper.
Author Response
We thank reviewer 3 for their comments, we believe that their comments have improved the quality of the manuscript.
Please see attachment for response to reviewers.

Round 2
Reviewer 3 Report
The authors are to be commended for a thorough response to this review. I am happy with all the altercations and feel the manuscript does a much better job in appealing to a broader audience and incorporation some background in fire ecology and management. At this point I only have one major comment and a few minor ones. I feel this paper should be accepted after these minor changes are implemented.
Now that the authors have clarified the reference data methodology, it has become clear that there has been no field-based validation of fire severity or vegetation class in this study. I feel like this is a caveat that must be addressed in the Discussion. I don’t know enough about the technical details of LiDAR assessment methods to say whether or not the lack of field validation affects the validity of the study, but I think it is necessary to have a paragraph explaining this caveat and convincing the reader that it is okay to have assessments of aerial images as a substitute for actual on the ground measurements as reference data.
Minor points:
L. 94 As stated in the previous review, Melaleuca alternifolia is not found in Tasmania. I can’t what Melaleuca species is the dominant understorey species at the cite, but the authors could surely use TASVEG to figure it out.
Table 1: The vegetation needs to be updated in this table.
L. 165 My interpretation of this based on the methods in the referenced papers is that this ‘assessment’ was a visual assessment of the ortho image. Is this correct? Please clarify
L. 510-516 I would suggest reminding the reader here of the resolution and classification accuracy values from this study, just to make the comparison a little easier.
L. 511 I feel like this sentence is incomplete. ‘While previous studies have demonstrated…” where’s the second half of this sentence?
Author Response
The authors thank the reviewer for the opportunity to further refine the manuscript.
Please see the attachment for specific comments addressing the reviewers concerns.
